# A Novel Score to Predict One-Year Mortality after Transcatheter Aortic Valve Replacement, Naples Prognostic Score

**DOI:** 10.3390/medicina59091666

**Published:** 2023-09-15

**Authors:** Zehra Güven Çetin, Ahmet Balun, Hülya Çiçekçioğlu, Bekir Demirtaş, Murat Mehmet Yiğitbaşı, Kerem Özbek, Mustafa Çetin

**Affiliations:** 1Cardiology Department, Ankara Bilkent City Hospital, 06800 Ankara, Turkey; drhulyac@gmail.com (H.Ç.); drkeremozbek@dr.com (K.Ö.); mdmustafacetin@yahoo.com (M.Ç.); 2Cardiology Department, Bandırma Onyedi Eylul University, 10200 Balıkesir, Turkey; ahmetbalun@gmail.com; 3Cardiology Department, Ankara Etlik City Hospital, 06170 Ankara, Turkey; bkrdemirtas@gmail.com; 4Sarıkamıs State Hospital, 36500 Kars, Turkey; nevresbey@gmail.com

**Keywords:** aortic stenosis, transcatheter aortic valve replacement, Naples Prognostic Score, mortality

## Abstract

*Background and Objectives*: Aortic stenosis (AS) is a widespread valvular disease in developed countries, primarily among the elderly. Transcatheter aortic valve replacement (TAVR) has become a viable alternative to aortic valve surgery for patients with severe AS who are deemed a high surgical risk or for whom the AS is found to be inoperable. Predicting outcomes after TAVR is essential. The Naples Prognostic Score (NPS) is a new scoring method that evaluates nutritional status and inflammation. Our study is aims to examine the relationship between the NPS and outcomes for patients receiving TAVR. *Material and Methods*: We conducted a retrospective study of 370 patients who underwent TAVR across three tertiary medical centres from March 2019 to March 2023. The patients were divided into two groups based on their NPS, namely, low (0, 1, and 2) and high (3 and 4). Our study is primarily aimed to determine the one-year mortality rate. *Results*: Within one year, the mortality rate for the entire group was 8.6%. Nonetheless, the low-NPS group had a rate of 5.0%, whereas the high-NPS group had a rate of 13%. The difference between the two groups was statistically significant, with a *p*-value of 0.06. *Conclusions*: Our results show that NPS is an independent predictor of one-year mortality in patients undergoing TAVR.

## 1. Introduction

Aortic stenosis (AS) is the most common acquired valvular disease in developed countries, with its prevalence escalating in tandem with advancing age [1]. The only proven treatment for severe AS is aortic valve replacement. However, advancing age, co-morbidities, frailty, previous surgeries, and cognitive dysfunctions complicate valvular surgery and the postoperative course. Thus, transcatheter aortic valve replacement (TAVR) has emerged as an alternative treatment to aortic valve surgery for patients with severe AS who are deemed a high surgical risk, or for whom the AS is found to be inoperable [2,3]. Numerous risk-scoring systems are available to predict the outcomes of surgical aortic valve replacement (SAVR) and TAVR. The Society of Thoracic Surgeons (STS) score and EuroSCOREII are well known risk-scoring systems routinely used to identify patients and whether they are suitable for surgery or percutaneous intervention, and to primarily predict the risk of surgical, in-hospital, and 30-day mortality [4,5]. The involved patient population has a high burden of co-morbidity, and solving the issue of frailty in the valvular problem may not always be enough to ensure a patient’s survival. Several factors, including cardiovascular status, extra-cardiac co-morbidities, and frailty, can significantly influence the long-term outcomes of TAVR beyond valvular disease [6]. Additional indicators in conjunction with surgical risk scores can enhance the prediction of outcomes and provide the highest quality of care possible after TAVR.

The Naples Prognostic Score (NPS) is a novel scoring system that consists of serum albumin and total cholesterol (TC) levels with neutrophil-to-lymphocyte ratio (NLR) and lymphocyte-to-monocyte ratio (LMR), which assesses the nutritional and inflammatory status of patients. The NPS was first validated as a prognostic factor for patients undergoing colorectal cancer surgery [7,8]. Recent studies have demonstrated that NPS is associated with mortality due to heart failure and short- and long-term outcomes in patients with ST-segment elevation myocardial infarction [9,10,11,12]. Since the outcomes after TAVR are closely linked to inflammation and nutritional status, we aimed to investigate the correlation between the NPS and prognosis after TAVR with a one-year follow-up in our study.

## 2. Materials and Methods

### 2.1. Study Design

For this study, we retrospectively analysed patients admitted to three distinct cardiology departments who were diagnosed with severe AS through echocardiographic means and were deemed to be eligible candidates for TAVR. From March 2019 to June 2022, we screened 426 patients who underwent TAVR for severe symptomatic AS; after applying exclusion criteria, 370 patients were enrolled in the study. Patient information, including baseline and procedural data, was retrieved from the digital database of the corresponding medical centre. In addition, follow-up data after the index hospitalisation were obtained from the national health database. The study did not include patients with severe anaemia (haemoglobin < 10 g/dL), a low platelet count (≤100,000/dL), chronic renal failure with a glomerular filtration rate (GFR) < 30 mL/min/1.73 m^2^, liver cirrhosis (Child–Pugh class C), active infection or sepsis, active immunological disease, active malignancies, or ongoing oncological therapies. Furthermore, patients who received valve-in-valve TAVR and emergency TAVR were excluded from the study.

### 2.2. Data Collection

All laboratory results were retrospectively obtained from digital databases. The blood analysis that was conducted on the day preceding the procedure as a part of the standard protocol was utilised to calculate NPS. NLR and LMR were calculated by dividing the absolute neutrophil count by the lymphocyte count, and by dividing the absolute lymphocyte count by the monocyte count, respectively. EuroSCORE II was calculated for each patient by the researchers during the data collection period. Information on TAVR procedural data and transthoracic echocardiography findings were retrieved from the digital database. In addition, we screened the digital database regarding procedural complications such as bleeding and emergency pacemaker requirements, as well as in terms of post-procedural stroke, acute coronary syndrome, acute kidney failure, bleeding requiring transfusion, and permanent pacemaker requirement during the index hospitalisation. Follow-up data were collected from the nationwide health and mortality database.

The study was approved by the Local Ethics Committee of Ankara City Hospital according to the Declaration of Helsinki. However, written informed consent was waived because of the retrospective and observational design of the study.

### 2.3. Scoring

The NPS has four components: 1. neutrophils/lymphocytes; 2. lymphocytes/monocytes; 3. total cholesterol level; and 4. serum albumin level. Figure 1 describes the calculation of the NPS. Patients with an NPS of 0, 1, and 2 were accepted as low-NPS, whereas patients with an NPS of 3 and 4 were accepted as high-NPS.

### 2.4. Outcomes

The primary outcome was all-cause mortality in the twelve months after TAVR. We have reviewed the patients’ digital records and the national database to assess all-cause mortality within a month, blood transfusions, permanent pacemaker requirement, post-procedure major bleeding, acute kidney injury, stroke, and transient ischemic attack during the procedure, as well as the length of the patient’s hospital stay. We searched the database for haemorrhagic/ischemic stroke, acute coronary syndrome, and permanent pacemaker implantation during follow-up after discharge. One-month mortality referred to any death within the first month after TAVI or during the index hospital stay. Acute kidney injury was defined as stages 2–4, while major bleeding is classified as types 2–4 according to VARC-3 criteria [13].

### 2.5. Statistical Analysis

SPSS software (version 17.0, SPSS Inc., Chicago, IL, USA) was selected for the statistical analyses. Parametric variables were given as mean ± standard deviation, non-parametric variables were given as median with 25–75th percentile, and categorical variables were given as percentages. Continuous variables were analysed using the Kolmogorov–Smirnov test for normal distribution. Depending on whether the continuous variables were normally distributed, differences among the groups were evaluated using Student’s *t*-test or the Mann–Whitney U test. Pearson chi-square and Fisher’s exact tests were performed to compare the differences among categorical variables in 2 × 2 tables. Univariate and multivariate regression analyses were performed to investigate the predictors of one-year, all-cause mortality in the study population. Variables significant at *p* < 0.10 in the univariate analysis were included in the multivariate logistic regression analysis to identify independent predictors of one-year, all-cause mortality. Univariate analyses of the variables are given in the Appendix A. Models 1 and 2 were created by adding the NPS to the baseline model as continuous and categorical variables. Model 1 included the NPS group, haemoglobin, chronic obstructive pulmonary disease (COPD), and atrial fibrillation (AF), and Model 2 included the NPS group, AF, haemoglobin, and EuroSCORE II. For the Kaplan–Meier survival analysis, patients were divided into two groups based on the Naples Prognostic Score, and survival changes were calculated using a log-rank test. A receiver operating characteristic (ROC) curve was applied to determine the cut-off EuroSCORE II and NPS values for predicting one-year mortality. We also adjusted mortality rates among groups based on creatinine, haemoglobin, and EuroSCORE II.

## 3. Results

We screened 426 patients, of which 56 were excluded due to exclusion criteria. Thus, 370 patients were included in the study. Of the 56 excluded patients, 29 were excluded because of chronic renal failure, 14 had severe anaemia, 5 had valve-in-valve TAVR, 3 had liver failure, 3 had rheumatoid arthritis, 1 had thrombocytopenia, and 1 was excluded due to missing data (Figure 2).

### 3.1. Baseline Characteristics

The average age of our cohort was 76.3 ± 7.0 years old. The baseline characteristics of the patients are detailed in Table 1. According to the New York Heart Association classification, the functional capacity of the majority of patients was poor, with 69.2% falling into classes 3–4. The low-NPS group and the high-NPS group were similar in terms of gender, previous heart surgeries, co-morbidities, and functional capacity (Table 1).

The median EuroSCORE II of our cohort was 5.9 (3.3–12.9), and the mean left ventricular ejection fraction was 50.4 ± 13. In the high-NPS group, the mean left ventricular ejection fraction (LVEF) was lower, and the EuroSCORE II was higher. The TAVR procedures were carried out through the transfemoral route, with self-expandable aortic prosthetic valves employed in 66% of cases. The number of patients who received self-expandable and balloon-expandable valves were similar in both groups. The mean haemoglobin level of the entire cohort was 11.9 mg/dL, but was lower in the high-NPS group. Additionally, the high-NPS group had a higher basal creatinine level. Regarding the NPS components, the NLR was higher, while the LMR, total cholesterol, and serum albumin levels were lower in the high-NPS group (Table 1). The entire group had an average total cholesterol level of 163.6 ± 42.9 mg/dL, with an average low-density lipoprotein (LDL) level of 94.3 ± 35.4 mg/dL, and 51.4% of patients were taking statins. In the low-NPS group, the average total cholesterol level was 179.2 ± 44.7 mg/dL, the average LDL level was 104.6 ± 39.8 mg/dL, and 54.2% of patients were taking statins. However, in the high-NPS group, the total cholesterol level was significantly lower at 144.9 ± 31.9 mg/dL, with an LDL level of 82.2 ± 24.5 mg/dL, and 47.9% of patients were taking statins. Despite the lower rate of statin use in the high-NPS group, there was no significant difference between the two groups. Unfortunately, we did not collect information on the statin dosage or patient compliance with the medication.

### 3.2. Post-Procedural Results

The incidence of major bleeding, post-procedural cerebrovascular events, acute coronary syndrome, urgent pacemaker implantation, and post-procedural acute kidney failure was comparable in both groups. Although the frequency of these complications was slightly higher in the high-NPS group, the difference was not statistically significant. During the index hospitalisation, the blood transfusion rate was 14.6%; it was slightly higher in the high-NPS group, but this difference was also not significant (Table 2). Fifteen of our patients died during the index hospitalisation. Of these patients, 11 were in the low-NPS group, and 4 were in the high-NPS group. The overall in-hospital mortality rate was 4.1% and statistically similar in both groups. During the initial month of follow-up, there were two deaths—one in each group—which was statistically similar (*p* = 0.902) (Table 2). The overall cohort had a mortality rate of 8.6% within one year. However, the low-NPS group had a rate of 5.0%, while the high-NPS group had a rate of 13%. The difference was statistically significant, with a *p*-value of 0.06 (Table 2). Both groups had similar incidences of cerebrovascular events, acute coronary syndromes, and permanent pacemaker implantation during the one-year follow-up after hospitalisation (Table 2).

Furthermore, both univariable and multivariable regression analysis models were conducted to identify the factors that predict one-year mortality. We included pre-existing AF, basal haemoglobin level, COPD, and high NPS in our regression models. Based on the multivariable analysis, pre-existing AF and high NPS were identified as independent predictors of one-year mortality with an odds ratio of 2.216 (1.047–4.689, *p* = 0.038) and 2.308 (1.035–5.146, *p* = 0.041), respectively (Table 3). Mortality rates were significantly higher in the high-NPS group when adjusted for creatinine, haemoglobin, and EuroSCORE II (4.9 ± 3.4 vs. 13.0 ± 8.9, *p* < 0.001) (Table 4). ROC analysis showed that a cut-off value of 3 for NPS predicted post-TAVI, one-year mortality with a sensitivity and specificity of 68.5% and 56.1%, respectively (area under the curve, 0.631; 95% CI, 0.527–0.734; *p* < 0.015), and EuroSCORE II predicted one-year mortality with a sensitivity and specificity of 59.4% and 59.5% (area under the curve, 0.641; 95% CI, 0.536–0.746), and a cut-off value of 7.42. (Figure 3).

The study involved a Kaplan–Meier survival analysis to determine the correlation between high NPS and mortality. The analysis revealed that a high NPS significantly impacted one-year mortality, with the difference becoming noticeable after six months of follow-up (log-rank *p* = 0.0071) (Figure 4).

## 4. Discussion

In the present study, we examined the prognostic value of pre-interventional NPS for patients who underwent TAVR. This is the first study evaluating the prognostic impact of NPS in TAVR recipients. Based on our results, NPS is a good predictor of mortality for TAVR recipients. According to Hoffmann et al.’s registry, a pre-existing inflammatory condition increases mortality rates and LV remodelling after TAVR [14]. Several studies have examined the predictive value of the NLR in patients who undergo TAVR [15,16,17,18]. All these studies have shown a significant correlation between pre-procedural NLR levels and clinical outcomes following TAVR. The most extensive study involved nearly 6000 patients who underwent either SAVR or TAVR in the PARTNER I, II, and S3 trials [18]. The study concluded that high baseline NLR levels were associated with higher rates of death and rehospitalisation at three years for SAVR and TAVR patients (58.4% vs. 41.0%, respectively; hazard ratio (HR): 1.39; confidence interval (CI): 95%, 1.18–1.63; *p* < 0.0001). Although previous studies have used different cut-off values for the NLR, the NLR cut-off value used for the NPS is similar to the high-tercile PARTNER registry.

AS, a chronic disease characterized by prolonged inflammation, can lead to reduced physical performance, loss of appetite, and a deterioration in nutritional status. This poses a particular concern for older patients with AS, as they often have reduced reserves and are more vulnerable to these adverse effects. Nutritional status has been extensively investigated in patients undergoing TAVR. Studies have shown that undernutrition has a negative impact on prognosis. The Geriatric Nutritional Risk Index (GNRI), Controlling Nutritional Status (CONUT), and Prognostic Nutritional Index (PNI) are well-known nutritional scoring systems that have been examined in TAVR patients [19,20,21,22,23]. Hypoalbuminemia has also been found to be strongly associated with mortality after TAVR, regardless of nutritional status. OCEAN-TAVI investigators found that patients with <3.5 mg/dL of serum albumin levels had higher mortality in short- and mid-term follow-ups after TAVR, particularly non-cardiac death [24]. In addition, VARC-2 defined hypoalbuminemia (serum albumin < 3.5 mg/dL) as a criterion for frailty, closely related to nutritional status [25]. Frailty is described as the decreasing physiological reserve of the patient, which is common in the elderly population [6,26]. In the current study, we did not evaluate frailty. Frailty encompasses a wide range of factors and cannot be solely attributed to nutritional status.

Hyperlipidaemia is a well-known risk factor for atherosclerosis and is suggested to be treated in patients with aortic stenosis [3]. In our entire cohort, half of the patients were on statin therapy. We acknowledge that evaluations of total cholesterol levels while on statin treatment may be questionable. For those with atherosclerotic disease, the use of statins is essential. Nevertheless, patients who were on statin therapy were similar in both groups, and interestingly, mean total cholesterol levels were below 180 mg/dL in both groups. Sudo et al. found that assessing nutritional status through total cholesterol, triglycerides, and body weight index (TCBI) was a reliable predictor of mortality three years post-TAVR. Approximately 68% of patients were on statins, which was significantly higher in the low-TCBI group. The authors concluded that the association between TCBI and all-cause mortality remained consistent regardless of statin use [27].

However, the NPS score did not seem related to major bleeding, in-hospital blood transfusion, acute stroke, and post-procedural acute kidney failure. We also found no difference between the high-NPS group and the low-NPS group regarding permanent pacemaker requirement after TAVR, which is a disturbing complication. A study conducted by Totaro et al. demonstrated that the NLR on the day of implantation is related to permanent pacemaker implantation (PPI) in patients undergoing TAVR without any conduction abnormalities in a cohort of nearly 180 patients with a rate of 13% and a cut-off value NLR ratio of >7.25 on TAVR day (AUC of 0.716; sensibility of 65%; and specificity of 73%, *p* = 0.003) [28]. The authors concluded that the preprocedural NLR and the NLR upon discharge were not related to PPI. In our study, the total pacemaker implantation rate during the index hospital stay was 13.1% and 14.2% for the high-NPS group, which was not significant. The pacemaker requirement at one-year follow-up was 2.2% for the overall cohort, which was similar between the groups. In this context, our results align with the study by Totaro et al. PPI is a troublesome complication after TAVR; it is closely related to the co-morbid status of the patient, the type of valve used, its generation, and the experience of the team performing the procedure. However, more extensive studies should be performed to understand the relationship between PPI and inflammation and nutritional status. Based on our research, pre-existing AF is also a predictor of mortality for TAVR recipients. This finding is consistent with the OCEAN TAVI registry produced by Hioki et al. [29]. In our cohort, almost one-third of patients had AF, with similar rates in both NPS groups. In univariable and multivariable analysis, mortality was higher in patients with AF (OR: 2.216; CI: 1.047–4.689; *p* = 0.038).

Previous studies have found that the success of TAVR is significantly linked to pre-existing levels of both inflammation and nutrition. The NPS scoring system is quite helpful in evaluating inflammatory and nutritional status. The study revealed that a high NPS score was an independent predictor of mortality in patients who underwent TAVR after one year, but not for in-hospital and one-month deaths. Our cohort’s mortality rate was 4.1% in-hospital, 0.5% in one month, and 8.6% after one year. The group with low NPS had a slightly higher in-hospital mortality rate, possibly due to procedural complications. The median EuroSCORE II value of the patient population was 5.9, 5.0 for the low-NPS group, and 7.4 for the high-NPS group, corresponding to a moderate-to-high surgical risk even for the low-NPS group. According to the Society of Thoracic Surgeons–American College of Cardiology Transcatheter Valve Therapy (STS-ACC TVT) Registry of Transcatheter Aortic Valve Replacement, the one-year mortality rate was 16.6% for high-risk patients and 8.3% for intermediate-risk patients [30]. In our study, one-year mortality was 8.6%, 5.0%, and 13% in the study population overall, the low-NPS group, and the high-NPS group, and the difference between the two groups was statistically significant (OR: 2.308; CI: 1.035–5.146; *p* = 0.041). After adjusting for creatinine, haemoglobin, and EuroSCORE II, mortality rates were significantly higher in the high-NPS group than in the other group. (4.9 ± 3.4 vs. 13.0 ± 8.9, *p* < 0.001). When EuroSCORE II was included in Model 2, the impact of NPS on mortality decreased [2.242 (0.991–5.074, *p* = 0.053)]. We cannot claim that NPS is as effective as EuroSCORE II in predicting mortality after TAVR. However, our study did not aim to compare the two scores. Nonetheless, NPS remains a valuable tool as it can be calculated using only routine blood tests. The scoring system does not include echocardiographic results, patient history, or clinical presentation, and does not need digital tools to calculate scores, unlike EuroSCORE II. Thus, this tool is affordable, easy to use, and provides quick results. On the other hand, the commonly employed surgical risk scores do not directly take into account nutritional condition and inflammation. However, they can offer a more complete evaluation and risk prediction when used alongside nutritional and inflammatory indices. Thus, the NPS could be employed to predict mortality following TAVR, in addition to well-known surgical risk scores. Since NPS simultaneously assesses both inflammation and nutrition, its usability is enhanced. Providing preprocedural information about patients’ nutritional status and inflammation may guide clinicians in follow-up. Patients with a high NPS may be followed more closely, and correcting undernutrition/malnutrition and limiting inflammation may help improve survival in TAVR recipients.

## 5. Conclusions

In conclusion, the NPS is a valuable tool for assessing inflammation and nutritional levels. Our study is innovative, as it delves into the significance of the NPS in patients who have undergone TAVR. Our results show that the NPS is a good predictor of one-year mortality in these individuals. We hope that our research will encourage further exploration in this field.

One of the major limitations of our study is its retrospective design; another limitation is the limited number of patients involved. Although we could have compared the post-procedural NPS results, it was not possible due to insufficient data availability. Unfortunately, we were unable to collect data on post-procedural and one-year follow-up functional capacity and rehospitalisation rates, which would have provided valuable insights into the benefits of TAVR for patients. It is essential to highlight that our results warrant further validation through prospective studies with larger sample sizes.

## Figures and Tables

**Figure 1 medicina-59-01666-f001:**
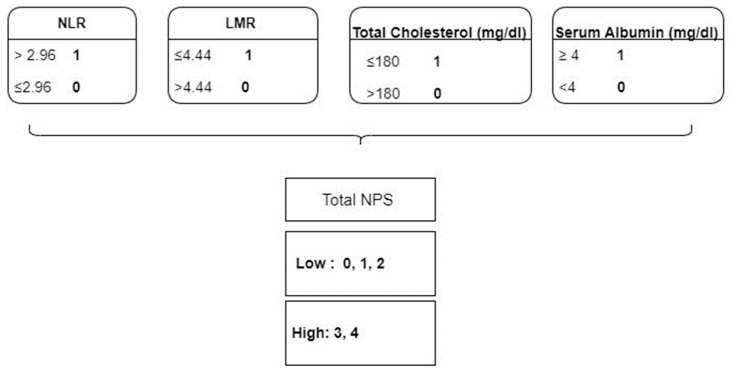
Formulation of Naples Prognostic Score; NLR, neutrophil-to-lymphocyte ratio; LMR, lymphocyte-to-monocyte ratio.

**Figure 2 medicina-59-01666-f002:**
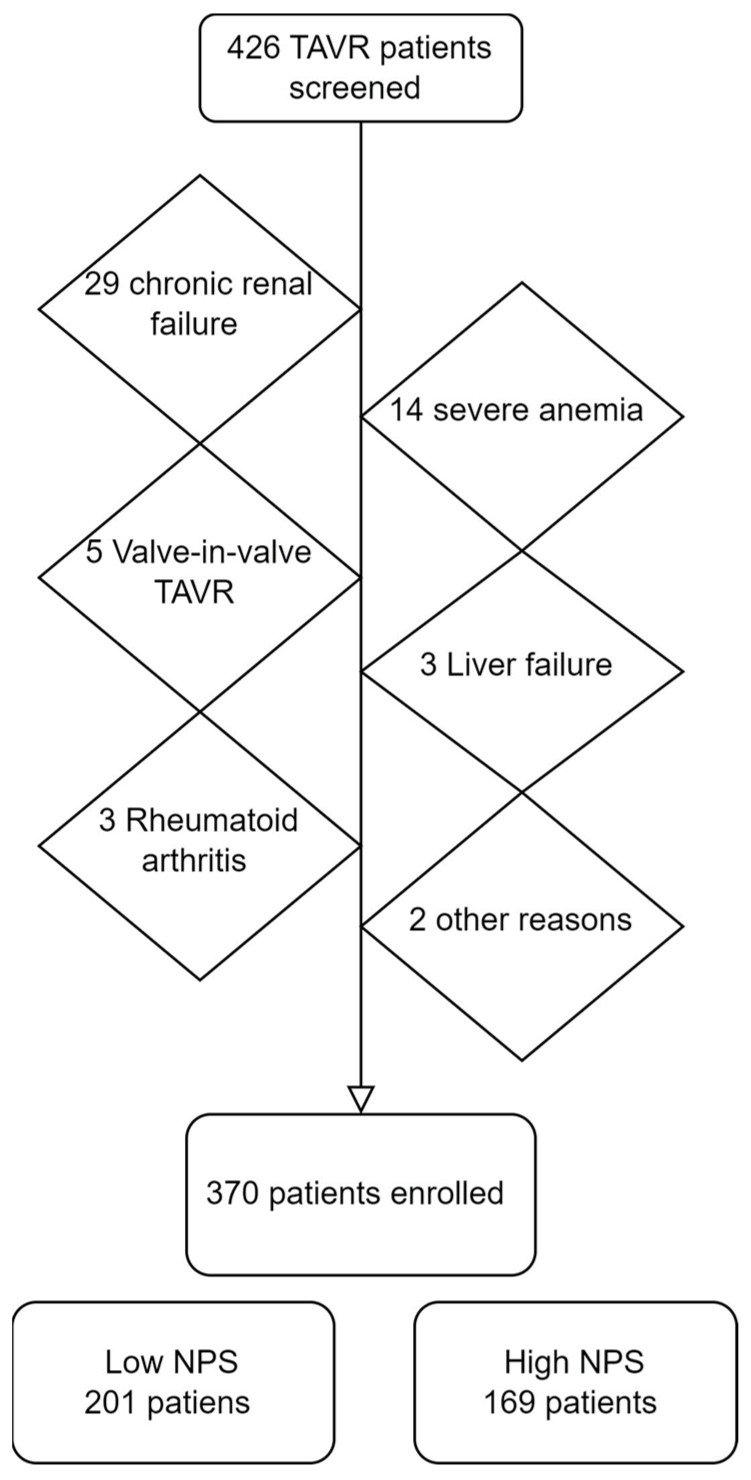
Study flow diagram.

**Figure 3 medicina-59-01666-f003:**
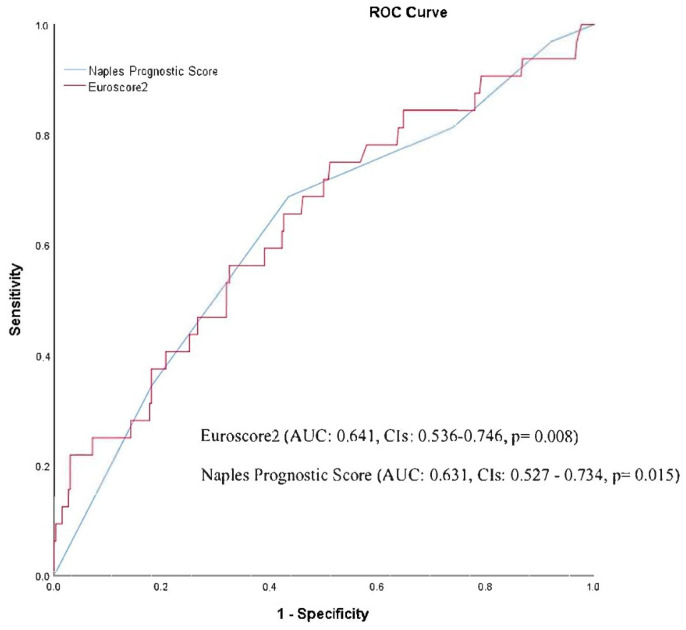
ROC analysis. The NPS predicted post-TAVI one-year mortality with a sensitivity and specificity of 68.5% and 56.1%, respectively, with a cut-off value of 3; EuroSCORE II predicted one-year mortality with a sensitivity and specificity of 59.4% and 59.5%, with a cut-off value of 7.4.

**Figure 4 medicina-59-01666-f004:**
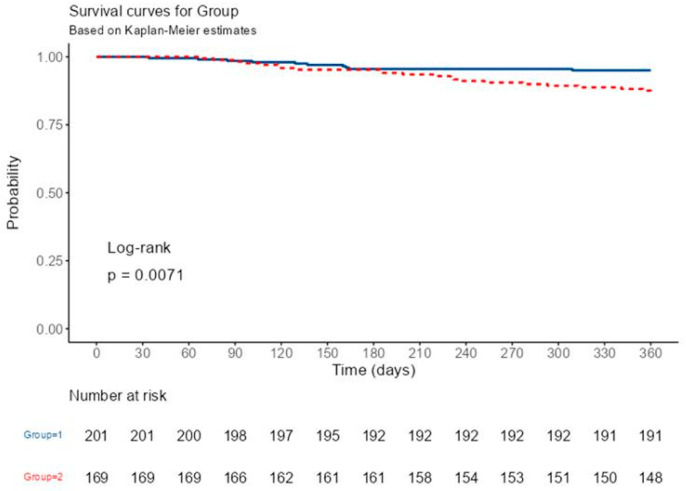
Kaplan–Meier survival curve for all-cause mortality in the follow-up period, according to Naples Prognostic Score (Blue line denotes Low NPS; dotted red line denotes High NPS).

**Table 1 medicina-59-01666-t001:** Demographic, clinical, and laboratory characteristics of the patients.

Variables	All (370)	Low NPS (201)	High NPS (169)	*p*
Age	76.3 ± 7.0	76.1 ± 6.9	76.4 ± 7.1	0.679
Male gender, n (%)	154 (41.6)	80 (39.8)	74 (43.8)	0.438
Hypertension, n (%)	329 (88.9)	183 (91.0)	146 (86.4)	0.155
Diabetes mellitus, n (%)	176 (47.6)	93 (46.3)	83 (49.1)	0.585
COPD, n (%)	98 (26.5)	46 (22.9)	52 (30.8)	0.087
Atrial fibrillation, n (%)	117 (31.6)	56 (27.9)	61 (36.1)	0.090
CVD, n (%)	31 (7.8)	13 (6.5)	18 (10.7)	0.148
Prior PCI, n (%)	131 (35.4)	66 (32.8)	65 (38.5)	0.260
Prior CABG, n (%)	67 (18.1)	39 (19.4)	28 (16.6)	0.481
NYHA 3–4, n (%)	256 (69.2)	136 (67.7)	120 (71.0)	0.488
Euroscore II	5.9 (3.3–12.9)	5.0 (2.9–11.0)	7.4 (3.9–14.1)	**0.004**
Aortic valve area, cm^2^	0.71 ± 0.17	0.72 ± 0.17	0.69 ± 0.17	0.183
LvEF, %	50.4 ± 13.1	52.4 ± 11.7	48.0 ± 14.2	**0.001**
WBC, ×1000/µL	7.4 ± 2.4	7.0 ± 1.9	7.9 ± 2.8	**<0.001**
Haemoglobin, g/dL	11.9 ± 1.7	12.2 ± 1.6	11.6 ± 1.8	**0.001**
Lymphocyte, (×1000/µL)	1.6 ± 0.6	1.9 ± 0.6	1.3 ± 0.5	**<0.001**
Neutrophil (×1000/µL)	5.0 ± 2.2	4.3 ± 1.6	5.9 ± 2.5	**<0.001**
Monocyte (×1000/µL)	0.5 ± 0.2	0.5 ± 0.2	0.5 ± 0.2	**0.007**
Platelet count (×1000/dL)	235.0 ± 82.2	230.8 ± 75.7	240.0 ± 89.3	0.282
Creatinine, mg/dL	1.03 ± 0.4	0.98 ± 0.35	1.09 ± 0.42	**0.007**
Total cholesterol, mg/dL	163.6 ± 42.9	179.2 ± 44.7	144.9 ± 31.9	**<0.001**
LDL, mg/dL	94.3 ± 35.4	104.6 ± 39.8	82.2 ± 24.5	**<0.001**
HDL, mg/dL	43.5 ± 12.1	45.7 ± 12.0	41.0 ± 11.7	**<0.001**
Total protein g/dL	64.8 ± 7.5	66.7 ± 7.3	62.5 ± 7.0	**<0.001**
Albumin g/dL	39.9 ± 5.3	41.8 ± 4.3	37.5 ± 5.5	**<0.001**
NLR	3.91 ± 3.20	2.47 ± 1.26	5.62 ± 3.90	**<0.001**
LMR	3.75 ± 2.43	4.69 ± 2.85	2.64 ± 1.04	**<0.001**
ASA	189 (51.1)	103 (51.2)	86 (50.9)	0.946
Clopidogrel	136 (36.8)	70 (34.8)	66 (39.1)	0.401
NOAC	101 (27.3)	53 (26.4)	48 (28.4)	0.662
Warfarin	25 (6.8)	12 (6.0)	13 (7.7)	0.511
Statin	190 (51.4)	109 (54.2)	81 (47.9)	0.227

COPD, chronic obstructive pulmonary disease; CVD, cerebrovascular disease; PCI, percutaneous coronary intervention; CABG, coronary artery by-pass graft; LvEF, left ventricular ejection fraction, WBC, white blood cell count; LDL, low-density lipoprotein; HDL, high-density lipoprotein; NLR; neutrophil-to-lymphocyte ratio; LMR, lymphocyte-to monocyte-ratio; ASA, acetylsalicylic acid; NOAC; new-generation oral anticoagulants.

**Table 2 medicina-59-01666-t002:** Procedural and postprocedural clinical outcomes of the groups.

Variables	All (370)	Low-NPS (201)	High-NPS (169)	*p*
**Procedural variables**				
Balloon-expandable, n (%)	122 (34.0)	64 (32.5)	58 (35.8)	0.509
**Post-procedural and in-hospital outcomes**				
Blood transfusion, n (%)	54 (14.6)	25 (12.5)	29 (17.2)	0.201
Major bleeding, n (%)	3 (0.8)	1 (0.5)	2 (1.2)	0.464
Pacemaker implantation, n (%)	48 (13.1)	24 (11.9)	24 (14.2)	0.519
CVE, n (%)	8 (2.2)	2(1.0)	6 (3.6)	0.092
AKI, n (%)	15 (4.1)	7 (3.5)	8 (4.7)	0.543
In-hospital death, n (%)	15 (4.1)	11 (5.5)	4 (2.4)	0.131
One-month death, n (%)	2 (0.5)	1 (0.5)	1 (0.6)	0.902
**Outcomes at one month to 12 months**				
Death, n (%)	32 (8.6)	10 (5.0)	22 (13.0)	**0.006**
Ischemic stroke, n (%)	13 (3.5)	8 (4.0)	5 (3.0)	0.595
Haemorrhagic stroke, n (%)	1 (0.3)	0 (0)	1 (0.6)	0.275
ACS, n (%)	1 (0.3)	1 (0.5)	0 (0)	0.359
Pacemaker implantation, n (%)	8 (2.2)	5 (2.5)	3 (1.8)	0.639

CVE, cerebrovascular event; AKI, acute kidney injury; ACS, acute coronary syndrome.

**Table 3 medicina-59-01666-t003:** Univariate and multivariate regression analysis for one-year mortality.

		Model-1	Model-2
	Univariate	Multivariate	Multivariate
	**OR** (95% confidence interval)	**OR** (95% confidence interval)	
Atrial fibrillation	2.347 (1.130–4.874, *p* = 0.022)	2.216 (1.047–4.689, ***p* = 0.038**)	2.003 (0.931–4.305, *p* = 0.075)
COPD	2.037 (0.965–4.299, *p* = 0.062)	2.049 (0.948–4.426, *p* = 0.068)	–
Haemoglobin	0.780 (0.626–0.974, *p* = 0.028)	0.823 (0.658–1.030, *p* = 0.089)	0.867 (0.686–1.095, *p* = 0.229)
NPS (3–4)	2.859 (1.313–6.222, *p* = 0.008)	2.308 (1.035–5.146, ***p* = 0.041**)	2.242 (0.991–5.074, *p* = 0.053)
EuroSCORE II	1.056 (1.026–1.086, *p* < 0.001)	–	1.050 (1.018–1.083, ***p* = 0.002**)

COPD, Chronic obstructive pulmonary disease; NPS, Naples Prognostic Score.

**Table 4 medicina-59-01666-t004:** Adjusted one-year mortality rates between groups.

Variable	All (370)	Low-NPS (201)	High-NPS (169)	*p*
Mortality (adjusted), % ± sd	8.6 ± 7.6	4.9 ± 3.4	13.0 ± 8.9	**<0.001**

sd: standard deviation.

## Data Availability

The data presented in this study are available upon request from the corresponding author. The data are not publicly available due to privacy and ethical restrictions.

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
