# Peer review of "A Novel Score to Predict One-Year Mortality after Transcatheter Aortic Valve Replacement, Naples Prognostic Score"

_medicina, 2023, doi:10.3390/medicina59091666_

Round 1

Reviewer 1 Report

The authors present Naples Prognostic Score as a predictor of mortality at 1 year following TAVR. While it is a great effort by the authors to present a scoring system to answer the question of mortality after TAVR, there are several questions that come up during the review of this paper. I would like to kindly ask the authors to comment on the following points.

1. The introduction desrcibes how TAVR is used for aortic stenosis and what the prevalence is.  Authors also descibe the indication of TAVR etc. However this information is redundant and has little to do with their research question. The main question in this paper is what predicts post TAVR mortality. While authors state that inflammation and nutritional status predicts long term mortality this important concept is ended in one line while the introduction spans over 2 big paragraphs. I would recommend introducing TAVR in just one line and present the main question in the introduction. 

2. Under data collection it is written that a skilled echocardiograher performed the echocardiograms. From my understanding this is a retrospective study, how was it determined that it was a skilled echocardiographer who performed the studies. The verbiage needs to be changed to reflect the retrospective nature of this study.

3. The entire method section is written in present tense, while this is a retrospective study and the verbiage needs to be corrected to reflect this is a retrospective study.

4. Labelling of groups should be done in methods section as it is difficult to follow which group is labelled as group 1 and which one is labelled as group 2.

5. The results section under baseline characteristics is long and redundant as most of the findings are desrcibed in the table. There is no need for repeating all the numbers in the baseline characteristics paragraph if those numbers are also written in the table. Authors should state important points in these baseline characteristics. Some of those important points are the statistical difference in LVEF, Euroscore, hemoglobin, and creatinine. And another important point is despite the use of statin the cholestrol difference in the two groups. Was there a difference in statin dose in these two groups. The point to highlight is despite the same use of statin in two different groups, the cholestrol levels are lower in high NPS group that may suggest nutritional status. Can this point be of improtance?

6. The authors state that there was a difference in mortality in the two groups. Please provide an adjusted mortality rates in the two groups when adjusted for differences in creatinine, hemoglobin, euroscore and other baseline characteristics if necessary.

7. Please share the univariate analysis is supplementary files.

8. The most important point in this sudy is discussion which needs to improve. The authors in their entire discussion state how their findings are similar to, or different from other studies. While this is important it does not need to take up the entire discussion. The main point of discussion is why do we need this scroring system and how will this scoring system change practice. Authors mention about using it for futility in one small line but does  a mortality rate of 13% at 1 year in high NPS score group call for futility when a higher percentage of these patients will die without treatment. What should be instead discussed is how can NPS score provide us with a different perspective of treatment post TAVR. Can we improve their nutritional status and inflammation status post TAVR so that we can improve outcomes. May be these patients need different kind of follow up. This score is helping us to find out patients who lack in nutritional aspect at the time of TAVR or patient who have high inflammatory burden, but what are we doing about that. How can NPS help us in this? Instead of describing the utility of NPS score, authors' main focus is to compare it to other modalities. Also the discussion is too long and redundant. It needs extensive revision.

9. Conclusion: The authors conclude their NPS score can predict futility, which is completely incorrect. How did authors come up with this conclusion? Was their a futility analysis? If 40% of untreated AS patients will die in 1 year, the mortality rate of 13% at one year in high NPS group is definitely not futile. 

10. Also, how were the two groups matched for type of valve used? this is not explained in the methods section?

Overall, while it is an interesting paper trying to present a new method to predict outcomes in patients undergoing TAVR, the authors need to provide as to why this should be used and how can we use these scores to improve outcomes? There is lot of redundant information in introduction and discussion which is definitely not warranted.  

There are a lot of grammatical mistakes

Author Response

Thanks to rewiever for the valuable comments. 

Reviewer 2 Report

Thank you for the opportunity to review this paper looking to correlate the NPS score with outcome following TAVI.

This is a well written manuscript which discusses the results well. My comments follow:

- Line 44 - you say 1 year mortality is 20% - i dont think that this is a contemporaneous figure and is much lower in my centre.

- I wonder if you can share your criteria for accepting a patient for TAVI. Is it based on age? surgical risk? etc, 

- Table 1 you have as the heading of the columns group 1 and 2 - should this not be low and high NPS?

- Pacemaker rate was 13.1%, that seems quite high - any comment on this?

- In your univariate/multivariate analysis could you include EuroSCORE 2 

- Could you perform ROC analysis to assess the relative snesitivity/specificity of your cut-off values for NPS - and perhaps compare this to the ROC analysis of EuroSCORE 2. My question is, what value does this NPS score add over and above well recognised scoring systems. Certainly the EusroSCORE was strongly associated with NPS high/low groups in Table 1

Author Response

Response to Reviewer 2 Comments

Comment 1: Line 44 - you say 1 year mortality is 20% - i dont think that this is a contemporaneous figure and is much lower in my centre.

We appreciate the reviewer's valuable comments. It has come to our attention that the mortality rate provided in the original manuscript referred to data from 2012. We have since reviewed the literature and updated the mortality rate to reflect the most current information available, which is up to 17% for high-risk patients, as stated in the STS-ACC TVT Registry of Transcatheter Aortic Valve Replacement published in 2020. We acknowledge that the older data in the introduction section was inadvertently overlooked. However, we did cite the registry and its results in the discussion section.

Comment 2: I wonder if you can share your criteria for accepting a patient for TAVI. Is it based on age? Surgical risk? etc, 

Our clinic's heart team is comprised of two heart surgeons, two interventional cardiologists, and an experienced anesthesiologist. They confidently make their decision on the most important factors, including surgical risk, co-morbidities, and the anatomy of the aortic annulus and aorta, ensuring the best possible outcome for our patients.

Comment 3: Table 1 you have as the heading of the columns group 1 and 2 - should this not be low and high NPS?

Response: According to reviewer’s comment we  have clarified the group names as ‘Low NPS and High NPS instead of Group 1 and Group 2

-Comment 4: Pacemaker rate was 13.1%, that seems quite high  any comment on this?

Our study focused on a cohort of high-risk patients with a high burden of co-morbidities, which increased the risk of procedural complications. We believe that the patient population we considered contributed to this risk. According to the STS-ACC TVT Registry, even lower-risk patients who require pacemakers pose a challenge, the overall pacemaker implantation rate were ranging from 16%-9,5%[1]. Factors such as the type of valve, the operator's experience, and the patient's clinic all influence the likelihood of pacemaker requirement. At our clinic, we have observed a decrease in the rate of pacemaker requirement with increasing experience. Totaro et al reported a 13% pacemaker implantation rate in their study, which is similar to our findings[2].

- Comment 5: In your univariate/multivariate analysis could you include EuroSCORE 2 

Response: We have created a new model that incorporates EuroSCORE II in our multivariate analysis.. The results are as follows:

Table 3. Univariate and Multivariate Regression Analysis for 1 year mortality

Model-1

Model-2

Univariate

Multivariate

Multivarite

OR (95% confidence interval)

OR (95% confidence intervaI)

Atrial fibrillation

2.347 (1.130-4.874, p=0.022)

2.216 (1.047-4.689, p=0.038)

2.003 (0.931-4.305, p=0.075)

COPD

2.037 (0.965-4.299, p=0.062)

2.049 (0.948-4.426, p=0.068)

-

Hemoglobin

0.780 (0.626-0.974, p=0.028)

  0.823 (0.658-1.030, p=0.089)

   0.867 (0.686-1.095, p=0.229)

NPS (3-4)

2.859 (1.313-6.222, p=0.008)

2.308 (1.035-5.146, p=0.041)

2.242 (0.991-5.074, p=0.053)

Euroscore2

1.056 (1.026-1.086, p<0.001)

-

1.050 (1.018-1.083, p=0.002)

Comment 6: Could you perform ROC analysis to assess the relative snesitivity/specificity of your cut-off values for NPS - and perhaps compare this to the ROC analysis of EuroSCORE 2. My question is, what value does this NPS score add over and above well recognised scoring systems. Certainly the EusroSCORE was strongly associated with NPS high/low groups in Table 1

Response: We conducted the ROC analysis as requested by the reviewer, which revealed that a cutoff value of 3 for NPS can predict one-year post-TAVI mortality, with a sensitivity and specificity of 68.5% and 56.1%, respectively (area under the curve, 0.631; 95% CI, 0.527–0.734; p < 0.015), and cutoff value of  7.42 for EuroSCORE II predict one-year mortality with a sensitivity and specificity of 59.4% and 59.5% ( area under the curve, 0.641; 95% CI, 0.536-0.746).

  1. Carroll, J.D.; Mack, M.J.; Vemulapalli, S.; Herrmann, H.C.; Gleason, T.G.; Hanzel, G.; Deeb, G.M.; Thourani, V.H.; Cohen, D.J.; Desai, N.; et al. STS-ACC TVT Registry of Transcatheter Aortic Valve Replacement. Journal of the American College of Cardiology 2020, 76, 2492-2516, doi:https://doi.org/10.1016/j.jacc.2020.09.595.
  2. Totaro, A.; Testa, G.; Calafiore, A.M.; Ienco, V.; Sacra, V.; Busti, A.; Pierro, A.; Sperlongano, S.; Golino, P.; Sacra, C. Neutrophil to lymphocyte ratio predicts permanent pacemaker implantation in TAVR patients. J Card Surg 2022, 37, 5095-5102, doi:10.1111/jocs.17212.

Round 2

Reviewer 1 Report

Thank you for giving the opportunity to re-review this manuscript. While authors made great effort to improve the manuscript, it still needs a lot of editing and improvement before it can be accepted. Authors have not adequately addressed the previous comments.

1. The introduction is still long and redundant. Please delete lines 31-34. Please delete lines 40-42. 

2. Keeping in mind that this is a retrospective study, under outcomes the authors still use wrong verbiage. It sounds like that this study was done as a prospective study and not a retrospective study. Please correct lines 118-125. For eg. "We also have recorded" and "We also monitored" etc.

3. Authors keep on mentioning that the type of valve were matched between two groups. Was there propensity matching done? How were they matched? I think authors are confusing the term "matching" here. Matching is a statistical term where patients were selected and matched based on a pre-determined critera. 

4. Similar matching term is also used for statins in the discussion.

5. Authors in the introduction mentioned that EUROSCORE is about perioperative mortality, but then in their results perform an ROC curve to show that a score of above 7.42 can predict one year mortality with a certain specificity and sensitivity. If that is the case, why is NPS score needed when EUROSCORE can predict that?

6. Discussion is very long and redundant. Currently the discussion is 13 paragraphs long, please reduce disussion to 5 paragraphs. Please discuss how NPS score can help us in selecting patients or improving patients' mortality.  One paragraph disussing NPS with other nutritional and inflammatory criterias should be fine. 

This manuscript needs extensive editing with respect to grammar and presentation. There are full stops before and after references at several places. Correct form of verbs have not been used. There is redundancy in this paper.

Extensive english editing required.

Author Response

Response to Reviewer 1 Comments

Point 1: The introduction is still long and redundant. Please delete lines 31-34. Please delete lines 40-42.

Response 1: Thank you once again to the reviewer for their valuable comments, contributions, and patience.

We have shortened the introduction section and deleted lines 31-34 and 40-42 as suggested by the reviewer.

Point 2: Keeping in mind that this is a retrospective study, under outcomes the authors still use wrong verbiage. It sounds like that this study was done as a prospective study and not a retrospective study. Please correct lines 118-125. For eg. "We also have recorded" and "We also monitored" etc.

Response 2: We have double-checked the data collection and outcomes sections and made the necessary corrections as per the reviewer's request.

Point 3. Authors keep on mentioning that the type of valve were matched between two groups. Was there propensity matching done? How were they matched? I think authors are confusing the term "matching" here. Matching is a statistical term where patients were selected and matched based on a pre-determined critera.

Response 3. Based on the reviewer's feedback, we have replaced the term 'matching' with 'similar in number'.

Point 4. Similar matching term is also used for statins in the discussion.

Response 4. Based on the reviewer's feedback, we have replaced the term 'matching' with 'similar in number'.

Point 5. Authors in the introduction mentioned that EUROSCORE is about perioperative mortality, but then in their results perform an ROC curve to show that a score of above 7.42 can predict one year mortality with a certain specificity and sensitivity. If that is the case, why is NPS score needed when EUROSCORE can predict that?

Response 5. We noted that EuroSCORE II, a widely used score, is a better predictor of TAVR mortality than NPS in both the introduction and discussion sections. We concluded that NPS and EuroSCORE II can be used together to predict mortality and improve survival in patients undergoing TAVR.

Point 6. Discussion is very long and redundant. Currently the discussion is 13 paragraphs long, please reduce disussion to 5 paragraphs. Please discuss how NPS score can help us in selecting patients or improving patients' mortality.  One paragraph disussing NPS with other nutritional and inflammatory criterias should be fine.

Response 6. Based on the reviewer's feedback, we have abbreviated the discussion section.

We are delighted to inform you that our article has undergone thorough professional editing and proofreading services. As a token of this, we would like to attach the certificate we have received.. We sincerely appreciate your continued support and interest in our work.

Reviewer 2 Report

Thank you for the opportunity to re-review the manuscript. You have addressed my comments well mostly.

I am still unconvinced about quoting mortality of 17% unless this is really qualified by saying that it is the 'high-risk' cohort of octagenarians. 17% does not reflect the standard mortality of this group of patients otherwise it would be rare to operate on patients over 80 which is not the case.

There has been significant improvement

Author Response

Response to Reviewer 2 Comments

Comment 1: I am still unconvinced about quoting mortality of 17% unless this is really qualified by saying that it is the 'high-risk' cohort of octagenarians. 17% does not reflect the standard mortality of this group of patients otherwise it would be rare to operate on patients over 80 which is not the case.

Response 1: Firstly, I would like to express gratitude to the reviewer for their valuable feedback and contributions on behalf of our team. Thank you for taking the time.

According to the recently published STS-ACC TVT Registry of Transcatheter Aortic Valve Replacement in 2021, high and extremely high surgical risk patients experienced a mortality rate ranging from 24% to 16.7% between 2011 and 2018. Meanwhile, those with intermediate surgical risk had a mortality rate of 8%, while those with low surgical risk had a mortality rate of approximately 8% [1]. The risk classification in this registry was based on the STS and TVT scores, which were median of 6.34 and 3.53, respectively, for high and extremely high-risk populations. The median age of this population was 82.

In our cohort, we assessed the surgical risk by EuroSCORE II, which was 5,9 for the whole population, and the median age was 76,3. The overall one-year mortality rate was 8.6%.

In the most recent version of the manuscript, we have removed the sentence "Although TAVR is a breakthrough in AS treatment and enables intervention for a significant number of patients, the one-year mortality rate for high-risk patients after TAVR is still up to 17%" to improve clarity.

  1. Carroll, J.D.; Mack, M.J.; Vemulapalli, S.; Herrmann, H.C.; Gleason, T.G.; Hanzel, G.; Deeb, G.M.; Thourani, V.H.; Cohen, D.J.; Desai, N.; et al. STS-ACC TVT Registry of Transcatheter Aortic Valve Replacement. Journal of the American College of Cardiology 2020, 76, 2492-2516, doi:https://doi.org/10.1016/j.jacc.2020.09.595.
